# Postoperative Morbidity after Dental Treatment under General Anesthesia in Children with and without Disabilities

**DOI:** 10.3390/medicina60040668

**Published:** 2024-04-19

**Authors:** Marija Šimunović-Erpušina, Danko Bakarčić, Odri Cicvarić, Luka Šimunović, Vlatka Sotošek, Dorotea Petrović, Nataša Ivančić Jokić

**Affiliations:** 1Department of Pediatric Dentistry, Faculty of Dental Medicine, University of Rijeka, 51000 Rijeka, Croatia; marija.simunovic.e@gmail.com (M.Š.-E.); odri.cicvaric@fdmri.uniri.hr (O.C.); dorotea.petrovic@uniri.hr (D.P.); natasa.ivancic.jokic@fdmri.uniri.hr (N.I.J.); 2Clinical Hospital Center Rijeka, 51000 Rijeka, Croatia; vlatkast@medri.uniri.hr; 3Department of Orthodontics, School of Dental Medicine, University of Zagreb, 10000 Zagreb, Croatia; lsimunovic@sfzg.hr; 4Department of Anesthesiology, Reanimatology, Emergency and Intensive Care Medicine, Faculty of Medicine, University of Rijeka, Braće Branchetta 20, 51000 Rijeka, Croatia; 5Department of Clinical Medical Sciences II, Faculty of Health Studies, University of Rijeka, Rijeka, Viktora Cara Emina 2, 51000 Rijeka, Croatia

**Keywords:** anesthesia, general, dental care, postoperative complications, developmental disabilities, dental care for children

## Abstract

*Background and Objectives:* General anesthesia induces reversible unconsciousness, eliminating sensation and enabling painless medical procedures. Vital for dental care, it addresses patients with medical conditions, those needing extensive procedures, and those unable to cooperate due to fear. Dental care for patients with disabilities is a significant unmet need, with studies showing increased oral disease prevalence. This research aims to analyze postoperative morbidity both in healthy and disabled children undergoing dental procedures under general anesthesia. *Materials and Methods:* This study involved 96 children aged 3 to 15 with dental caries. Two groups were formed: the control group (CTL) (52.94%) included healthy patients requiring general anesthesia due to unsatisfactory cooperation, and the other group included children with physical or intellectual disabilities (CD) (47.96%). Postoperative complications were monitored 1 h after the procedure and 1, 3, 7, and 14 days after the procedure by contacting parents/caregivers by phone. The intensity of postoperative pain was assessed using the Wong–Baker faces pain rating scale. General anesthesia was conducted following a standardized protocol for dental procedures. *Results:* CTL patients complained more often about postoperative pain 1 h after the procedure (*p* = 0.03). One day after the procedure, oral bleeding (*p* = 0.04), fever (*p* = 0.009), and diarrhea (*p* = 0.037) occurred more often in CD. In the same period, sore throat appeared more often in CTL (*p* = 0.036). Three days after the dental treatment, there were statistically significant increases in the occurrence of agitation (*p* = 0.043) and constipation (*p* = 0.043) in CD. Seven days later, CD demonstrated a higher occurrence of agitation (*p* = 0.037). According to the Wong–Baker faces pain rating scale, CTL experienced more frequent and intense pain. *Conclusions:* CD more frequently reported complications like oral bleeding, fever, and diarrhea. In contrast, the CTL group more often reported pain-related symptoms. The conclusion underscores the need for a pain control protocol for CD and emphasizes the importance of frequent monitoring to prevent unwanted consequences during tooth restoration under general anesthesia.

## 1. Introduction

General anesthesia is a fundamental element in modern medical practice, inducing temporary unconsciousness and eliminating sensation, thus enabling painless completion of various medical procedures [1]. Within dentistry, its application has become indispensable for ensuring optimal care for patients whose conditions cannot be effectively managed through alternative means [2]. The American Academy of Pediatric Dentistry advocates for the judicious use of general anesthesia, particularly in cases where patients cannot cooperate due to factors such as psychological immaturity, mental or physical disabilities, or when local anesthesia proves insufficient. Moreover, it is recommended for individuals exhibiting fear, anxiety, or non-cooperation, as well as for preverbal or non-communicative children and those necessitating complex surgical dental interventions [3]. Early childhood caries remains a common dental problem and is a significant public health concern, and tooth restoration under general anesthesia notably improves the patient’s oral-health-related quality of life. Thoroughly assessing a patient’s suitability for this procedure is crucial, considering expenses and potential risks linked to general anesthesia [4,5].

Providing adequate dental care for patients with disabilities represents a significant yet often overlooked healthcare imperative. These individuals exhibit a heightened susceptibility to oral diseases, primarily due to the inherent challenges in administering essential interventions such as caries treatment, pulp therapy, and extractions [4,6]. The complex nature of disabilities patients can manifest in difficulties comprehending unfamiliar environments and procedures, leading to expressions of distress or disruptive behavior, which can further impede effective communication and interaction with dental staff [7]. Consequently, the administration of dental care utilizing local anesthesia presents notable challenges, particularly in cases involving cognitive impairments or behavioral disorders. Thus, dental rehabilitation conducted under general anesthesia emerges as a preferable course of action in certain circumstances [6]. Furthermore, children who have had negative experiences in medical or dental office may exhibit uncooperative behavior during subsequent visits, underscoring the increasing demand for the utilization of general anesthesia in both pediatric and disabled patient populations [8].

Dental procedures under general anesthesia offer the advantage of completing treatment in a single visit, minimizing discomfort for patients, parents/caregivers, and dentists. Although there are numerous advantages of dental restoration under general anesthesia, a child may experience some postoperative complications, such as postoperative pain, swelling, weariness, agitation, mastication problems, drowsiness, oral bleeding, fever, cough, sore throat, nausea, constipation, epistaxis, vomiting, excitement, and diarrhea [9,10,11]. Additionally, dentists face limited follow-up opportunities with such patients after the procedure, and research findings regarding postoperative issues vary in severity [12].

The aim of this research is to analyze the frequency of postoperative morbidity and intensity of postoperative pain in healthy children and in children with physical or intellectual disabilities after dental procedures under general anesthesia performed at the Dental Clinic of Clinical Hospital Center Rijeka, Croatia.

## 2. Materials and Methods

Our data were reported using the STROBE checklist.

### 2.1. Patients

This prospective observation study involved children aged 3 to 15 who had received treatment at the Department of Pediatric Dentistry of the Clinical Hospital Center Rijeka, who had been diagnosed with dental caries, and who were recommended for dental restoration under general anesthesia.

This study was approved by the Ethics Committee of the Clinical Hospital Center Rijeka (No: 2170-29-02/1-21-2, 17 March 2021), the Ethics Committee of the Faculty of Dental Medicine, University of Rijeka (No: 2170-57-006-21-1, 27 April 2021), and the Ethics Committee for Biomedical Research of the Faculty of Medicine, University of Rijeka (No: 2170-24-04-3-21-7, 26 October 2021), in accordance with the Declaration of Helsinki [13].

The parents/caregivers of all patients signed informed consent forms before participation. This study was conducted from April 2021 to December 2022 and included 96 patients divided into two groups. The control group (CTL) consisted of 49 (51.04%) healthy but anxious and phobic patients, and it was not possible to establish a satisfactory level of cooperation to carry out the dental procedure in the dental office. The second group consisted of 47 (48.96%) children with physical or intellectual disabilities (CD) who underwent dental treatment under general anesthesia at the Dental Clinic of the Clinical Hospital Center Rijeka, Croatia.

The inclusion criteria for rehabilitation under general anesthesia were children experiencing dental caries, uncooperative patients, medically compromised patients, patients with developmental disabilities, and patients with a presence of pain or discomfort as a result of untreated caries. Exclusion criteria included patients whose parents/caregivers refused to participate in the study, patients who underwent oral surgery on soft tissues (frenectomies), and patients in whom procedure caused extensive destruction of bone tissue (retained and impacted teeth). The exclusion criteria employed during the patient selection process are illustrated in Figure 1.

The diagnoses of the CD patients was as follows: autism in 24.48% (24); varying degrees of intellectual disability with or without epilepsy in 8.16% (8); malformation syndromes in 4.08% (4); cerebral palsy in 3.06% (3); ADHD in 2.04% (2); Down syndrome in 2.04% (2). The rest of the subjects (4.08%) were patients (4) with tetralogy of Fallot, fetal alcohol syndrome, epilepsy, and one oncological patient.

### 2.2. Anesthesia and Dental Procedures

All children who were referred for dental procedures under general anesthesia underwent an anesthesiologist’s examination before the procedure and were referred for the procedure with the specialist’s consent. The administration of general anesthesia followed a standardized protocol to minimize potential variables. Patients fasted and abstained from water for at least six hours before the procedure. Upon entering the operating room, an anesthesiologist set up standard devices for patient monitoring.

Anesthesia induction involved the inhalation of sevoflurane (Sevorane; AbbVie, Campoverde di Aprilia, Italy). Intravenous access was established once the eyelash reflex disappeared, and medications including propofol (Propofol-Lipuro; Braun, Frankfurt, Germany) and sufentanil (Sufentanil; Renaudin, Itxassou, France) were administered. Nasotracheal intubation was performed on the majority of patients, but if this was not possible, they were intubated orotracheally. Maintenance of anesthesia was achieved through a combination of sevoflurane and propofol. To prevent the aspiration of secretions and dental materials, a throat gauze patch was applied to all patients before surgery.

Dental procedures were carried out by two pediatric dentists following the American Academy of Pediatric Dentistry guidelines. All the necessary procedures were performed in one visit (conservative restorations, endodontic treatments, tooth extraction). The treatment plan was more radical in order to reduce the possibility of complications and avoid the repeated need for another tooth restoration under general anesthesia. In other words, teeth with good prognosis were restored or endodontically treated, while those with bad or questionable prognosis were extracted.

Following surgery, tracheal extubation occurred once patients regained consciousness and exhibited re-established respiration, swallowing, and cough reflexes. Subsequently, patients were transferred to the post-anesthesia care unit, where continuous monitoring and oxygen supplementation persisted until they met the discharge criteria. Before discharge, anesthetists provided parents/caregivers with standardized post-anesthesia guidance, and dentists informed them about performed dental procedures, oral hygiene instructions, and analgesia. Medical records, anesthesia details, and vital signs were documented.

The presence of postoperative morbidity and the intensity of postoperative pain were monitored 1 h after the procedure, and 1, 3, 7, and 14 days after the procedure by contacting parents/caregivers by phone. The occurrence of postoperative complications was monitored, which included postoperative pain, swelling, weariness, agitation, mastication problems, drowsiness, oral bleeding, fever, cough, sore throat, nausea, constipation, epistaxis, vomiting, excitement, and diarrhea. The intensity of postoperative pain was determined using the Wong–Baker face pain scale, which was given to the parent/caregiver and explained during the examination of the child [14]. The scale is composed of pictures of six faces, each of which represents the patient: no pain (0), mild pain (2), moderate pain (4), severe pain (6), very severe pain (8), and worst pain (10). There were numbers on the scale that the parent/caregiver could use to rate the pain based on the face the child chose 1 h after the procedure, and 1, 3, 7, and 14 days after the procedure (Figure 2).

### 2.3. Statistical Analysis

A sample size calculation was conducted to determine the necessary number of participants. Under the assumption of a one-tailed *t*-test, with an effect size (d) of 0.5, an alpha error probability of 0.05, and a desired power (1-β error probability) of 0.8, the analysis suggested a minimum total sample size of 102 patients, with 51 in each of the two independent groups. The sample sizes for the two groups were slightly lower than anticipated, with 49 participants in group 1 and 47 in group 2. However, a post hoc power analysis, reflecting the actual conditions of the study, revealed different insights. Post hoc analysis indicated that the achieved power (1-β error probability) of the study was approximately 85.13%, surpassing the initially targeted power.

In this analysis, SPSS Statistics 29.0.1.0 (IBM, Armonk, NY, USA) software served as the primary tool for statistical examination. The first step involved assessing the normality of the data distribution, which was carried out using QQ plots and the Shapiro–Wilk test. For the comparison of categorical data, two tests were employed: the Pearson Chi-squared test and Fisher’s exact test. As none of the continuous data demonstrated normality, the Mann–Whitney U test was utilized for group comparisons. Throughout the analysis, a significance level was set, where *p*-values lower than 0.05 were considered statistically significant.

## 3. Results

### 3.1. Demographic, General Anesthesia, and Dental Procedures Data

Demographic, general anesthesia, and dental procedures data are presented in Table 1. These include sex distribution, age, premedication practices, local anesthesia usage, the duration of anesthesia, the method of intubation used, and the number of extracted, extracted and sutured, and restored teeth during the dental procedures.

The analysis showed that there were no significant differences between the two groups in most aspects, except the age of the patients.

CTL were significantly younger (6, IQR 4.5–8) than CD (8, IQR 6–10) (*p* = 0.002). The median of the DMFT index was 0 (IQR 0–4) and did not differ between the groups (*p* = 0.11). On the other hand, the patients in CTL (9, 7–13) had a statistically significantly higher dft index compared with CD (7, 0–11) (*p* = 0.014).

### 3.2. Postoperative Morbidity

Postoperative morbidity data at five time points are shown in Table 2. Postoperative morbidity included postoperative pain, swelling, fatigue, agitation, chewing problems, drowsiness, oral bleeding, fever, cough, sore throat, nausea, constipation, epistaxis, vomiting, arousal, and diarrhea.

One hour following the procedure, postoperative pain was more prevalent in CTL compared with CD, and this difference was statistically significant with a *p*-value of 0.03.

On the first day after the procedure, CD exhibited a higher occurrence of postoperative oral bleeding compared with CTL, and this difference was statistically significant with *p* = 0.04. Furthermore, on the day following the procedure, CTL exhibited a 6.1% frequency of fever, while CD showed a higher incidence at 25.5%, indicating a statistically significant difference with a *p*-value of 0.009. Additionally, diarrhea was absent in CTL (0%), but it manifested in 8.5% of patients in CD, and this distinction was statistically significant with a *p*-value of 0.037. Within 1 day post procedure, sore throat was experienced in 24.5% of CTL patients, whereas in CD, only 8.5% experienced sore throat, demonstrating a statistically significant difference with a *p*-value of 0.036.

Three days after the dental treatment under general anesthesia in CD, a statistically significant increase in the occurrence of agitation (*p* = 0.043) and constipation (*p* = 0.043) was observed.

Seven days after the dental treatment under general anesthesia, CD demonstrated a higher occurrence of agitation (*p* = 0.037) in comparison with CTL.

In CTL, mastication problems and pain were the most common 14 days after the procedure, while mastication problems, cough, and agitation were the most prevalent in CD. These results have no statistical significance.

Pain intensity was determined using the Wong–Baker face pain scale (WB pain scale), and the results are shown in Figure 3.

One hour after the procedure, according to the WB pain scale, pain was absent in CTL in 81.60% of cases. Pain was absent in 91.50% of patients in CD.

One day after the procedure, mild pain was present in 42.90% of the CTL patients. Mild pain was present in 29.50% of patients in CD, while 50% reported no pain.

Three days after the procedure, mild pain occurred in 24.50% of patients in CTL, and 73.50% of patients reported no pain. Mild pain occurred in 17.80% of patients in CD, while 77.80% reported no pain.

Seven days after the procedure, CTL reported mild pain in 6.10%, while 93.90% reported no pain. Mild pain occurred in 6.70% of patients in CD, while 91.10% reported no pain.

Fourteen days after the procedure, 95.90% of patients in CTL and 97.80% of patients in CD reported no pain.

Based on the findings of this research, CTL reported higher levels of pain intensity, although there was no statistical significance in all monitored time intervals.

## 4. Discussion

The primary aim of this study is to assess the occurrence of postoperative complications and the level of postoperative pain among both healthy individuals and those with physical and intellectual impairments following dental restoration performed under general anesthesia. Following tooth rehabilitation under general anesthesia, restoration of the masticatory surface morphology is achieved, thereby reinstating the chewing function of decay-affected teeth. This procedure also contributes to the restoration of oral microflora balance, leading to a reduction in caries risk and the likelihood of developing dental anxiety. Nevertheless, children undergoing procedures under general anesthesia may encounter various postoperative issues, including pain, swelling, fatigue, restlessness, chewing difficulties, drowsiness, oral bleeding, fever, cough, sore throat, nausea, constipation, nosebleeds, vomiting, excitement, and diarrhea [9].

This study describes the postoperative morbidity that occurred in children after dental procedures under general anesthesia in the Clinical Hospital Center Rijeka, Croatia. The frequency of their appearance in the group of children with and without developmental difficulties was investigated in order to determine their significance on the patient’s health. Most patients experienced at least one symptom of postoperative morbidity related to the dental procedure or general anesthesia, but most resolved after three days. This finding is consistent with the ones reported in previous research [11,15,16,17].

Comparing these studies poses a challenge due to the presence of variables involved, including variations in the medical and cognitive condition of subjects, variances in caregiver socioeconomic backgrounds, differences in pain assessment tools, inconsistencies in the standardization of general anesthesia and dental procedures, variations in the quantity and types of dental interventions performed, and disparities in the utilization of postoperative pain relief medications [12].

According to this research, in both groups, males are more often referred for tooth restoration under general anesthesia. Following other studies, boys are more likely to receive dental treatment under general anesthesia. The reason for this disproportion cannot be said exactly [2,8,18,19,20,21,22]. However, a possible reason for this finding could be slower psychological maturation leading to an unwillingness to cooperate satisfactorily under conditions of local anesthesia [11]. In contrast, some studies report more frequent dental restorations under general anesthesia in women [2].

Patients in CTL were younger than CD, and CTL has a statistically significantly higher dft index compared with CD. This finding can be explained by the fact that the dft index is decreased in primary teeth while DMFT index is increased in permanent teeth. The decrease in dft scores is linked to the diminished number of primary teeth over time, primarily due to exfoliation [23]. A possible reason for the presence of older patients in the CD group may be the difficult access to dental services among such patients [24]. Davis et al. conducted a study in which 75% of children with difficulties did not receive dental services within a single year [25]. Similarly, Desai et al. conducted research on dental care in children with difficulties. Their dental needs remained unaddressed in 41% of cases despite needing basic dental treatment [26]. Inadequate utilization of dental services appears to be influenced by a variety of factors. These encompass physical challenges such as issues with transportation, economic barriers, including the limited coverage of services in public dental care programs that lead to caregivers incurring out-of-pocket expenses, behavioral difficulties such as children’s reluctance to cooperate during dental appointments, and a shortage of general dental practitioners who are willing to serve individuals with physical or intellectual difficulties [24]. Dental healthcare tends to be a lower priority for the CD group, such as those with autism spectrum disorder, who experience a significantly higher rate of unmet oral healthcare needs compared with healthy patients [27]. Parents/caregivers usually postpone seeking dental care because they focus on the general medical needs of their child [28].

Based on this research, when considering potential confounding variables such as gender, premedication practices, duration of anesthesia, intubation method, dental procedures, and local anesthesia, no significant differences were observed between the two groups in these aspects (Table 1).

All parameters of postoperative morbidity in CTL and CD at five time points are shown in Table 2.

One hour after the procedure, CTL patients complained more often about the existence of pain in 18.4% of cases, unlike CD (4.3%). Zhang et al. demonstrated that within the initial 72 h post-surgery, 94.86% of patients experienced some form of postoperative discomfort, with postoperative pain being the most prevalent (62.7%). Prior research indicates a variability in the occurrence of postoperative pain, ranging from 36% to 95%. Notable variations among the studies can be ascribed to discrepancies in participant demographics, types of dental procedures, the utilization of local anesthesia, and diverse pain assessment tools. Pain, being a subjective experience, exhibits individual variability. Self-reported pain remains the gold standard for assessment, yet it is necessary to consider the restricted capacity of young patients to articulate their sensations [9]. The CD group mostly consisted of patients on the autism spectrum who have limited verbal communication skills. Therefore, it is necessary to pay attention to such patients who cannot verbalize the existence of pain. Due to the complex nature of recognizing and addressing pain in children with disabilities, it is common for pain to be overlooked or not adequately managed. The International Association for the Study of Pain emphasizes that the absence of verbal communication does not eliminate the potential for an individual to undergo pain and require suitable pain relief measures [29]. Pain behaviors encompass the visible signs and expressions exhibited by a person in pain. The act of observing pain behaviors is regarded as a credible method for assessing pain in individuals who are incapable of self-reporting. These signs include different vocalizations, painful facial expressions, inability to calm such a patient, a patient who withdraws and seeks comfort, lack of or too much sleep, increased movement of the arms and legs compared with the usual state, stiffness of the extremities, clenching of the hands, arching of the back, tachycardia, sweating, tremors, atypical behavior and facial expressions, laughter, breath holding, and self-injurious behaviors [29].

In the 1 day after the procedure, CD more often complained about the existence of oral bleeding (24.4%), unlike CTL (8.2%). In CD, oral bleeding may be more visible due to swallowing dysfunction and motor disabilities related to neurological conditions [30]. However, Enever et al. reported oral bleeding during the first 72 h in one CD and two patients without difficulties [31]. According to Zhang, 30.27% of children had postoperative oral bleeding within the initial 24 h following the procedure [9].

Additionally, one day after the procedure, fever (25.5%) and diarrhea (8.5%) appeared more often in CD. However, in the same period, CTL was more likely to complain of sore throat (24.5%). In accordance with our findings, CD was more prone to develop postoperative fever (25.5%). The appearance of fever is a symptom that worries parents. The frequency of fever after dental general anesthesia ranged from 1% to 50% in previous studies [9]. According to Cantekin et al., it appeared in 18% of cases [32]. Similar to this research, Farsi showed the appearance of fever in 21% of patients [11]. Some of the factors that can lead to an increase in body temperature include bacteremia, tissue trauma, dehydration, pulmonary atelectasis, drugs such as atropine, and environmental conditions such as the temperature in the operating room or the patient’s covering during surgery. Clinical studies have established a connection between postoperative fever and dehydration after general anesthesia [33]. Zhang et al. state that low nutritional levels increase the risk of postoperative fever. Furthermore, a higher nutritional status appeared to have a protective effect on postoperative fever. High nutritional status was associated with a lower likelihood of developing postoperative fever due to bacteremia, soft and hard tissue damage, and dehydration. The researchers concluded that the majority of fever cases occurred between 24 and 72 h postoperatively. None of the children in the study developed a severe infection. This could potentially be explained by the prolonged preoperative fasting and water deprivation, irrespective of age and inability to eat postoperatively. Those factors may have contributed to postoperative dehydration and fever in the pediatric patients [9].

Additionally, in our research, one day after the treatment, diarrhea was absent in CTL (0%), but it manifested in 8.5% of patients in CD. Zhang et al. state that diarrhea is the least common postoperative complication after dental treatments under general anesthesia (2.7% of participants). It is said that children’s gastrointestinal tract may be in a state of stress due to long periods of preoperative fasting [9]. According to Cantekin et al., diarrhea was present in 5.1% of patients [32].

CTL reported sore throat almost three times more often than CD one day after the procedure (24.5%). Farsi et al. reported sore throat in 34.4% of patients on the first day [11]. Nasotracheal intubation may irritate or injure the lining of the respiratory system, causing throat swelling, coughing, and epistaxis [9]. The reason for more frequent reports of sore throat in the healthy CTL group of patients can be communication difficulties and failure to report any pain in CD [29]. Furthermore, postoperative complaints of sore throat may arise from the traumatic nature of intubation, especially when multiple attempts are made, and from the use of a double throat pack by pediatric dentists. It is advisable to focus efforts on gentle manipulation of throat tissues during intubation to minimize the likelihood of postoperative sore throat issues [11].

Based on the findings of this study, agitation (12.8%) and constipation (12.8%) manifested in CD three days post-surgery and are statistically significant. Additionally, seven days after the procedure, a higher incidence of agitation (8.5%) was observed in CD.

Agitation is a prevalent complication associated with general anesthesia. The incidence of agitation following dental general anesthesia has been reported to range from 26% to 76%. Notably, Zhang et al. found that agitation is present in 45.14% of cases within the first 72 h [9]. A meta-analysis conducted by Costi et al. [34] encompassing 14,045 children revealed that those administered sevoflurane as an inhalation anesthetic during general anesthesia exhibited postoperative irritability compared with the halothane group. Children may experience postoperative agitation due to their heightened sensitivity to environmental shifts, prolonged preoperative fasting leading to hunger and thirst, as well as the emotional stress of separation from their parents [9]. Furthermore, in CD due to insufficient communication, such unusual and irritable behavior may be indicative of pain [29].

More frequent findings of constipation in CD (12.8%) can be explained by the fact that the child’s gastrointestinal tract is in a state of stress due to a long fasting period. According to Zhang et al., constipation was present in 9.19% of patients [9]. However, according to Mathew and Jeevanandan, constipation was present in 2% of cases 24 h after the procedure [35].

According to the WB pain scale in this research, in CTL, pain was more intense and patients reported it more often compared with CD. More intense pain was reported in CTL, but it was not statistically significant in all observed time intervals. In both groups, more intense pain was reported after 1 day. In CD, the reported pain level decreased to the level of the 1st hour on the 7th day. On the 14th day, it fell below the level that was reported after 1 h. In healthy patients, though, the reported pain on the 7th day was already below the reported pain in the 1st hour. Hu et al. reported that the peak pain intensity was observed in the post-anesthesia recovery room in healthy children, specifically one hour after surgery. Subsequently, pain ratings exhibited a gradual decline at the intervals of 1 day, 3 days, 7 days, and 14 days postoperatively [12]. The limitations of this study are the small number of patients in each group and the variability of the examined group. Further studies with a larger number of patients in both groups are needed with a more homogeneous research group. Furthermore, a limitation of this study is the age of the patients and the reliance on the parents’ perceptions of postoperative complications. As parents possess profound knowledge of their children, this approach currently stands out as the most reliable method. Moreover, this study is limited by its dependence on phone-based assessments, requiring trust in the accuracy of symptom reporting by parents.

By reviewing the recent literature, we identified that this field of dental anesthesia is poorly researched, which prompted this research. According to our research, the main contribution of this study is the realization that CD subjects do not have poorer oral health compared with healthy children. They rarely complain of subjective pain-related symptoms, while the occurrence of oral bleeding, fever, diarrhea, constipation, and agitation is more common compared with healthy patients. Given their primary diagnosis, a different approach to CD is necessary. Considering that patients with CD more frequently report oral bleeding within the initial 24 h, it is advisable to conduct a follow-up examination the day after the procedure. To prevent fever in CD, it is recommended to avoid prolonged periods of fasting and water deprivation. Additionally, to reduce the risk of diarrhea and constipation, the duration of fasting and water deprivation should be optimized based on the nutritional status of CD individuals. Moreover, the onset of agitation may be linked to the choice of anesthesia, extended fasting periods, separation from parents, and unfamiliar surroundings; however, it can also serve as an indicator of the existence of pain. The International Association for the Study of Pain emphasizes that the absence of verbal communication does not eliminate the potential for an individual to undergo pain and require suitable pain relief measures. Hence, it is crucial to implement effective pain management measures for all CD patients.

## 5. Conclusions

Based on the results of this research, the CD group more often reported the occurrence of oral bleeding, fever, diarrhea, agitation, and constipation, while CTL patients experienced the symptoms of pain and sore throat more frequently. These subjective symptoms are related to the existence of pain. According to the Wong–Baker face pain scale, pain was more frequent and more intense in the healthy patients. To prevent unpleasant outcomes associated with dental procedures under general anesthesia that occurred in both groups, it is imperative to precisely outline the treatment protocol.

## Figures and Tables

**Figure 1 medicina-60-00668-f001:**
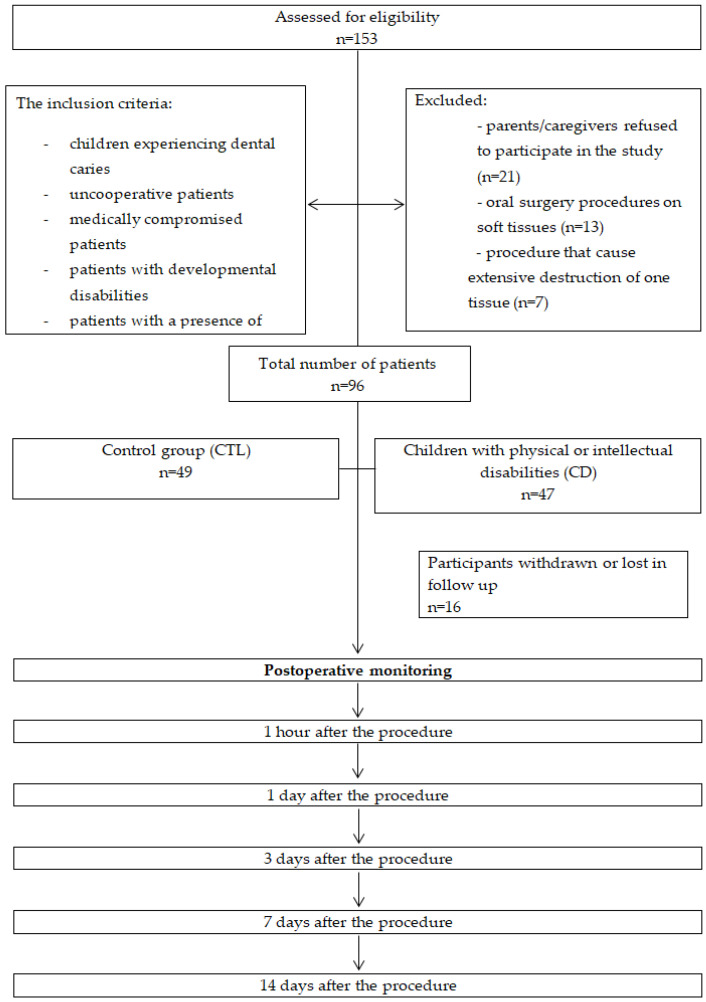
Flow chart of the study.

**Figure 2 medicina-60-00668-f002:**
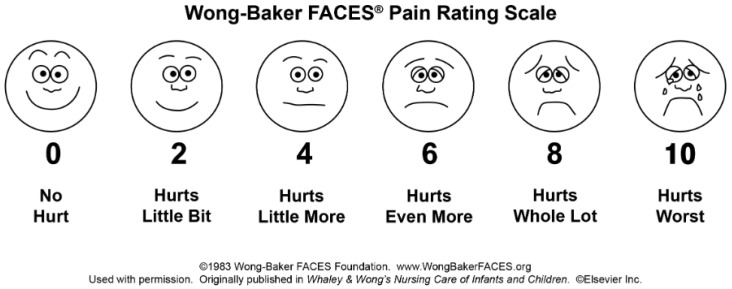
Wong–Baker FACES pain rating scale. Used with permission.

**Figure 3 medicina-60-00668-f003:**
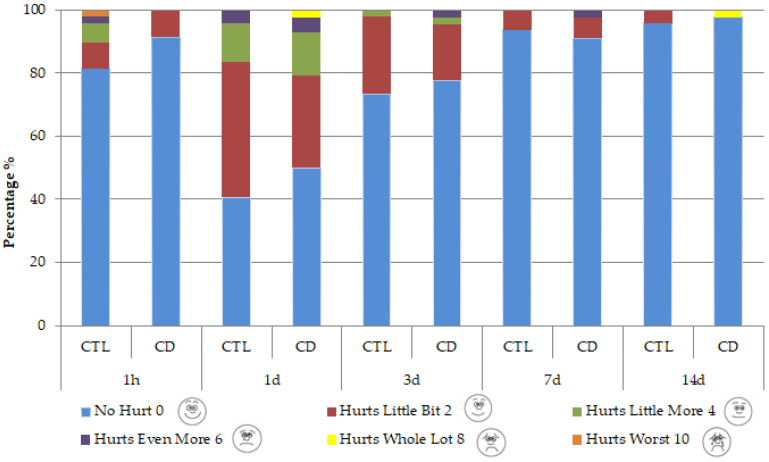
WB pain scale at five time points: 1 h, 1 day, 3 days, 7 days, and 14 days after the procedure.

**Table 1 medicina-60-00668-t001:** Demographic, general anesthesia, and dental procedure data.

Variable Name		Control Group (CTL)		Children with Physical or Intellectual Disabilities (CD)		
		*N*/Total (%)	Median (IQR)	*N*/Total (%)	Median (IQR)	*p*-Value
Gender	Male	27/49 (55.1%)		34/47 (72.3%)		0.079 *
	Female	22/49 (44.9%)		13/47 (27.7%)		
Age			6.0 (4.5–8.0)		8.0 (6.0–10.0)	0.002 ‡
Use of local anesthetic	Yes	47/48(97.9%)		42/46 (91.3%)		0.153 †
	No	1/48 (2.1%)		4/46 (8.7%)		
Total duration of anesthesia (min)			90 (75–103)		90 (75–105)	0.96 ‡
Premedication	Yes	32/43 (74.4%)		29/33 (87.9%)		0.144 †
	No	11/43 (25.6%)		4/33 (12.1%)		
Intubation	Nasotracheal	30/34 (88.2%)		40/43 (93.0%)		0.468 †
	Orotracheal	4/34 (11.8%)		3/43 (7.0%)		
Dental procedures	Teeth restored		4 (2–6)		4 (2–6)	0.91 ‡
	Teeth extracted		2 (0–5)		1 (0–4)	0.066 ‡
	Teeth extracted and sutured		0 (0–6)		1 (0–4)	0.96 ‡

* Chi-squared test; † Fisher’s exact test; ‡ Mann–Whitney U test.

**Table 2 medicina-60-00668-t002:** Parameters of postoperative morbidity at five time points: 1 h, 1 day, 3 days, 7 days, and 14 days after the surgery.

	1 h			1 d			3 d			7 d			14 d		
	CTL	CD	*p*	CTL	CD	*p*	CTL	CD	*p*	CTL	CD	*p*	CTL	CD	*p*
Pain	9 (18.4%)	2 (4.3%)	0.030 *	29 (59.2%)	20 (44.4%)	0.153 *	13 (26.5%)	9 (20.0%)	0.455 *	3 (6.1%)	3 (6.4%)	0.958 *	2 (4.1%)	1 (2.2%)	0.524 †
Swelling	1 (2.0%)	1 (2.1%)	0.742 †	16 (32.7%)	12 (25.5%)	0.443 *	7 (14.3%)	8 (17.0%)	0.712 *	0 (0.0%)	1 (2.1%)	0.490 †	0 (0.0%)	0 (0.0%)	
Weariness	6 (12.2%)	2 (4.3%)	0.157 *	6 (12.2%)	13 (27.7%)	0.058 *	2 (4.1%)	1 (2.1%)	0.516 †	1 (2.0%)	4 (8.5%)	0.154 *	0 (0.0%)	0 (0.0%)	
Agitation	4 (8.2%)	1 (2.1%)	0.183 *	5 (10.2%)	6 (12.8%)	0.694 *	1 (2.0%)	6 (12.8%)	0.043 *	0 (0.0%)	4 (8.5%)	0.037 *	0 (0.0%)	2 (4.3%)	0.237 †
Mastication problems	0 (0.0%)	1 (2.1%)	0.490 †	28 (57.1%)	18 (38.3%)	0.065 *	19 (38.8%)	17 (36.2%)	0.792 *	12 (24.5%)	8 (17.0%)	0.368 *	5 (10.2%)	5 (10.6%)	0.944 *
Drowsiness	27 (55.1%)	32 (68.1%)	0.191 *	11 (22.4%)	12 (25.5%)	0.724 *	1 (2.0%)	2 (4.3%)	0.484 †	1 (2.0%)	1 (2.1%)	0.742 †	0 (0.0%)	0 (0.0%)	
Oral bleeding	22 (44.9%)	20 (42.6%)	0.817 *	4 (8.2%)	11 (23.4%)	0.040 *	2 (4.1%)	3 (6.4%)	0.612 *	1 (2.0%)	0 (0.0%)	0.510 †	1 (2.0%)	0 (0.0%)	0.510 †
Fever	0 (0.0%)	0 (0.0%)		3 (6.1%)	12 (25.5%)	0.009 *	0 (0.0%)	2 (4.3%)	0.237 †	0 (0.0%)	1 (2.1%)	0.490 †	0 (0.0%)	1 (2.1%)	0.490 †
Cough	2 (4.2%)	1 (2.1%)	0.508 †	9 (18.4%)	6 (12.8%)	0.450 *	1 (2.0%)	4 (8.5%)	0.154 *	0 (0.0%)	2 (4.3%)	0.237 †	0 (0.0%)	2 (4.3%)	0.237 †
Sore throat	4 (8.3%)	1 (2.1%)	0.176 *	12 (24.5%)	4 (8.5%)	0.036 *	5 (10.2%)	3 (6.4%)	0.498 *	1 (2.0%)	0 (0.0%)	0.510 †	0 (0.0%)	0 (0.0%)	
Nausea	1 (2.1%)	0 (0.0%)	0.505 †	1 (2.0%)	2 (4.3%)	0.484 †	1 (2.0%)	0 (0.0%)	0.510 †	0 (0.0%)	0 (0.0%)		0 (0.0%)	0 (0.0%)	
Constipation	0 (0.0%)	0 (0.0%)		6 (12.2%)	7 (14.9%)	0.705 *	1 (2.0%	6 (12.8%)	0.043 *	1 (2.0%)	2 (4.3%)	0.484 †	0 (0.0%)	1 (2.1%)	0.490 †
Epistaxis	2 (4.2%)	7 (14.9%)	0.074 *	1 (2.0%)	3 (6.4%)	0.287 *	0 (0.0%)	1 (2.1%)	0.490 †	0 (0.0%)	0 (0.0%)		0 (0.0%)	0 (0.0%)	
Vomiting	1 (2.1%)	2 (4.3%)	0.492 †	2 (4.1%)	5 (10.6%)	0.217 *	0 (0.0%)	1 (2.1%)	0.490 †	0 (0.0%)	0 (0.0%)		0 (0.0%)	0 (0.0%)	
Excitement	0 (0.0%)	3 (6.4%)	0.075 *	0 (0.0%)	3 (6.4%)	0.072 *	0 (0.0%)	0 (0.0%)		0 (0.0%)	1 (2.1%)	0.490 †	0 (0.0%)	0 (0.0%)	
Diarrhea	0 (0.0%)	0 (0.0%)		0 (0.0%)	4 (8.5%)	0.037 *	2 (4.1%)	2 (4.3%)	0.676 †	1 (2.0%)	2 (4.3%)	0.484 †	0 (0.0%)	0 (0.0%)	

* Chi-squared test; † Fisher’s exact test.

## Data Availability

All the data used in this study are available on request from the corresponding author.

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
