# Peer review of "Postoperative Morbidity after Dental Treatment under General Anesthesia in Children with and without Disabilities"

_medicina, 2024, doi:10.3390/medicina60040668_

Round 1

Reviewer 1 Report (Previous Reviewer 1)

Comments and Suggestions for Authors

The manuscript improved substantially compared to the initial version. The doubts were resolved and in general the authors adequately addressed the observations and recommendations made previously, so it can be accepted. There are some typographical errors in the language that will surely be addressed in the editing stage.

Comments on the Quality of English Language

Minor editing of English language required

Author Response

Dear,

thank you for the review and positive evaluation of the article.

Best regards,

Authors

Reviewer 2 Report (Previous Reviewer 2)

Comments and Suggestions for Authors

Thank you for the significant changes.

Author Response

Dear,

thank you for the review and positive evaluation of the article.

Best regards,

Authors

Reviewer 3 Report (Previous Reviewer 3)

Comments and Suggestions for Authors

This article "Postoperative morbidity after dental treatment under general 2 anesthesia in children with and without disabilities" was improved significantly upon reviewing the third revision. 

The written, flow, scientific presentation of the data are clear and easy to follow.

The major flaw is a lack of strobe check list. The whole paper did not mention that they reported the study following strobe check list. But this paper does have most part of the items listed in the stroke check list. 

Von Elm, E., Altman, D. G., Egger, M., Pocock, S. J., Gøtzsche, P. C., & Vandenbroucke, J. P. (2007). The Strengthening the Reporting of Observational Studies in Epidemiology (STROBE) statement: guidelines for reporting observational studies. The lancet, 370(9596), 1453-1457.

The reviewer cannot accept this article being published without following the strobe guideline and uploading the strobe checklist with the submission.

Minor issues:

1 Method: Add reference for Declaration of Helsinki

World Medical Association. (2013). World Medical Association Declaration of Helsinki: ethical principles for medical research involving human subjects. Jama, 310(20), 2191-2194.

2 Method, where is your reference for Wong-Baker FACES Pain Scale?

Wong, D. and Baker, C.: Pain in children: comparison of assessment scales, Pediatric Nursing, 14(1):9-17, 1988.

3 There are a few comments I mentioned in 1st and 2nd round of review that they responded they revised according, but in fact it was unchanged in the manuscript, which are minor issues. Please go back to revision comments for the flow chart, and table 1.

Author Response

Dear,

thank you for the review, we have corrected everything acording to your comments and below are listed answers on your comments:

Comment No 1: This article "Postoperative morbidity after dental treatment under general anesthesia in children with and without disabilities" was improved significantly upon reviewing the third revision.

Answer: Thank you very much for your review.

Comment No 2: The written, flow, scientific presentation of the data are clear and easy to follow.

Answer: Thank you very much for the observation.

Comment No 3: The major flaw is a lack of strobe check list. The whole paper did not mention that they reported the study following strobe check list. But this paper does have most part of the items listed in the stroke check list.

Answer: Thank you for your valuable observation. The STROBE check list is added.

Minor issues:

Comment No 4: 1 Method: Add reference for Declaration of Helsinki

Answer: Thank you for your suggestion. A reference to the Declaration of Helsinki was added.

Comment No 5: 2 Method, where is your reference for Wong-Baker FACES Pain Scale?

Answer: Thank you for your suggestion. A reference to the Wong-Baker faces Pain Scale was added.

Comment No 6: There are a few comments I mentioned in 1st and 2nd round of review that they responded they revised according, but in fact it was unchanged in the manuscript, which are minor issues. Please go back to revision comments for the flow chart, and table 1.

Answer: Thank you for your suggestion. Flow chart and Table 1. are changed.

Best regards

Authors

This manuscript is a resubmission of an earlier submission. The following is a list of the peer review reports and author responses from that submission.

Round 1

Reviewer 1 Report

Comments and Suggestions for Authors

Postoperative morbidity after dental procedures in general anesthesia in healthy patients and in patients with physical or intellectual disabilities

·       In the introduction it should be clear to readers what the justification or reason for a report of this nature is. Emphasize it.

·       Eliminate this phrase from the objective section and include and analyze it in the discussion, if applicable and of interest: "By reviewing the recent literature, we identified that this field of dental anesthesia is poorly researched, which prompted this research."

·       As this is a randomized controlled clinical trial, it is essential to adhere to the CONSORT criteria, so authors are invited to complete these criteria and report them in their manuscript.

·       It is not clear why it is a randomized study, what was randomized? The study design must be specified. It does not appear to be a randomized clinical study.

·       In the case of a clinical study, how was the sample size calculated?

·       This sentence is confusing: “The treatment plan was more radical in order to reduce the possibility of complications and avoid the repeated need for tooth restoration. In one surgery, all necessary procedures were performed (conservative restorations, endodontic treatments, tooth extraction)”. It seems that the type of treatments included was not controlled or defined. The title talks about dental procedures, which ones were included?

·       This is important to try to relate the main complications reported.

·       Consider other limitations of the study and not only the number of patients included.

·       Was the time of the procedure controlled, was it the same between the study groups? In the discussion this is mentioned, but its implications are not discussed or inferred.

·       In the group of patients with disabilities, how can we trust what the patient reports, for example the pain variable?

·       Are the reported complications not typical of general anesthesia? Discuss this factor.

·       The first paragraph of the discussion is confusing, it should refer to the main objective of the study.

·       The proposed hypothesis is not taken up in the discussion.

·       The discussion should be more integrated, several paragraphs appear isolated and are only described and not discussed or compared.

·       What is the main contribution of the study?

·       The discussion must be redone and be clear about the main points to be addressed and how they relate to each other. Limit conclusions to the results of the study and do not make other inferences.

·       It was very difficult to follow the manuscript.

Comments on the Quality of English Language

 Moderate editing of English language required

Author Response

Dear,

please see the attachment, it contains point-by-point response to your comments.

Best regards

Danko Bakarčić

Reviewer 2 Report

Comments and Suggestions for Authors

This is a multiple cohort study comparing the postoperative morbidity (measured through assessing different parameters e.g. pain, swelling, fever, sore throat, ….. etc.) as well as postoperative pain intensity in healthy children and children with disabilities indicated for dental care under general anesthesia (GA). The topic is quite important to patients, their parents/care givers, and clinicians. The study is in general well-designed and conducted, however, the study analysis requires further revision as to analysis itself and data presentation. The STROBE reporting guideline should be more diligently followed. The manuscript can also benefit from language revision. Following are the comments:

Abstract

-          The results section requires more specification with the augmentation with p-values for the outcomes that showed statistically significant differences to allow drawing more correct conclusions.

Materials and methods

-          Was sample size calculation undergone?!

-          Were confounders identified and adjusted for?!

Results

-          A flow diagram is recommended showing the number of patients per group and the data about missing data if any.

-          Presenting the baseline characteristics of patients in both groups is recommended.

-          Table 1: was the presented data frequencies or percentages? This needs to be clarified. Highlighting the specific instances (which outcome parameter and at which timepoint) the data that showed statistically significant differences between groups by using symbols or font variations or any other means is necessary.

-          “One hour after the procedure, drowsiness and oral bleeding are most common in Group 1, while the same complications: drowsiness and oral bleeding occur most frequently in Group 2.” The statement is unclear. Kindly rephrase.

-          Was there any statistically significant difference between groups regarding pain intensity?!

Comments on the Quality of English Language

Minor revision required.

Author Response

(The authors gave the same response as above.)

Reviewer 3 Report

Comments and Suggestions for Authors

Author Response

(The authors gave the same response as above.)

Round 2

Reviewer 1 Report

Comments and Suggestions for Authors

The manuscript improved and clarified some doubts raised in the initial review, especially the design, which is very important in the conclusions and inferences made, since inferring from an experimental design is not the same as an observational one. The included flow chart is very useful to understand the study. The methodological details were attended to.

There are some observations that in my opinion are not completely resolved and must be addressed:

1. The introduction remains fragmented, 2 or 3 paragraphs with sufficient clarity may be enough to describe the context in which the study is carried out and justified.

2. Expand and discuss study limitations; for example, now by recognizing that it is not an experimental design, this is not addressed as a limitation, being very important and even more so that a hypothesis has been raised.

3. The authors must try to better integrate the discussion, expand ideas and compare with other related studies is essential.

4. The main contribution of the study should be better reflected.

5. I want to insist that the conclusion must be limited to the objective. Inferences are included in the discussion.

6. Table 2 must be integrated, described and referred to in the results.

Comments on the Quality of English Language

Minor editing of English language required.

Author Response

Dear,

we provide point-by-point response your comments uploaded as a Word file. Please see the attachment.

Best regards,

Danko bakarčić

Reviewer 2 Report

Comments and Suggestions for Authors

Thank you for the significant changes.

Comments on the Quality of English Language

Language requires minor revision.

Author Response

(The authors gave the same response as above.)

Reviewer 3 Report

Comments and Suggestions for Authors

Title: Postoperative Morbidity after Dental Procedures in General Anesthesia in Healthy Patients and in Patients with Physical or Intellectual Disabilities

Second Version.

Upon reviewing the second version, I can see the author’s effort and many improvements for this article, especially on the figures and tables. The reviewer would like to acknowledge their hard work.

However, there are still many writings that could be improved significantly. The major flaw will still be the way of reporting the study as well as the unclear novelty and clinical meaning. It is finally clear that this is an observational study with two cohorts: Healthy children, and children with disabilities who need dental care under general anesthesia. It was confusing previously when the article stated it is a randomized trial. Now please note that you should carefully follow the STROBE guidelines for cohort studies when reporting your data. Please remember to attach the filled STROBE checklist with any future submission.

https://www.strobe-statement.org/

Please check the following comments:

Major

1.      The reviewers really need to see your checklist. It is hard to judge the work without the filled checklist, it is very time consuming when we review your work, we have to follow the list and check every corner to find the clue.  If you already followed the guidelines checklist, please attach your filled form.

2.      Your flow chart needs LOTS of improvement.

a.      What are your inclusion and exclusion criteria for each cohort? Please add details.

b.      The diagram showed sample sizes of 58 and 54 in healthy and disability group, but your final result is 49 and 47 respectively. It is a blurry figure that I can hardly read and found the withdraw information. We don’t need the details of withdraw occurring from post-op to day 14, please simply  report that 16 participants withdrawn or lost in follow up.

c.      Why don’t you follow the STROBE FLOW DIAGRAM for cohort studies. It will be so much clearer if you follow it.  The reviewer feels you tried to follow it but did not really follow it, it is confusing. Why not ask your mentor for feedback, if they understand it, the reviewer and readers can understand it.

3.      The figure could be merged into a panel, so that you have more space to show more nice figures to visualize your result. For example, you only showed pain, and oral bleeding in figures 4 and 5, why don’t you make similar figures for fever, diarrhea, sore throat, constipation, agitation, mastication difficulties, Make it into one figure panel of 6-9 small figures. This allows you to show more interesting impact of comprehensive dental procedure under General anesthesia has impact on patient’s oral cavity, pain, emotion, body immune system.

4.      Reviewer really does not see the value of fig 2 and 3, what is the point here?  Do you really need to show this figure. Also, these two figures can be merged together.

5.      What is the novelty of this work? Does it add any value to clinical practice?

Minor:

1.      Title: I almost bit my tongue when I read out this title. Please carefully consider revising your title, avoid using “in” all the time. Here is the suggested title, and feel free to revise it base on your understanding: Post-operative Complications/Symptoms following comprehensive dental procedures under general anesthesia in children (or grade-schoolers) with and without physical or intellectual disabilities.

2.      Group name: It is hard to follow group 1 and group 2 in the article. Consider name them differently with abbreviations. Such as Children with disabilities (CD) and control (CTL) group. Change it in your writing, table, and figures.

3.      Table 1 needs to be improved.

a.      For the left column where you explain the variables, please put is as VaribaleName (format of report), eg: Gender [Female/Total (%)], CD, 22/49(44.9%), CTL 13/47 (27.7%), p-value 0.079 *; Age [Median(IQR)], Use of local anesthesia [Yes/Total (%)]; Incubation route [nasotracheal/total(%)], premedication blabla.

b.      So, table 1 will be demographic information and dental anesthesia and procedure information according to your table title.

It is not surgical data; restoration is not surgery. You can name it dental procedure. Please group your anesthesia related together and group the dental procedures together. Add a new title as a new row.

c.      Table 1 Please add name of statistical analysis you used at the table legend at the bottom. Eg * note proportional data, and method, † note continuous data and another method (eg t-test or Mann-Whitney U test), ‡ note another different data and statistical methods, the rest that is not marked is another statistical analysis.  This also applies to your table 2 for the results.

3. Figure 2-3 can be merged, 4-5 can be merged together.

4. For any figures, please mark the statistical significance with * like what you did on figure 4&5.

5. Figure color:

a. Great job using blue and purple for the green-red color-blind reader.

b. Figure 5 is so much better, but color is confusing, the severity from 0 to 10 is not marked from lighter to darker color. Red (2) or purple (6) are misleading because these colors look more severe than blue. You also used very similar blue color for 0 and 10. Please seriously consider changing the colors.

Your discussion first paragraph is much better. What is the novelty and clinical significance?

Author Response

(The authors gave the same response as above.)
